# Lineage marker synchrony in hematopoietic genealogies refutes the PU.1/GATA1 toggle switch paradigm

Michael K. Strasser[1,4], Philipp S. Hoppe[2], Dirk Loeffler[2], Konstantinos D. Kokkaliaris [2], Timm Schroeder[2], Fabian J. Theis [1,3] & Carsten Marr [1]

Molecular regulation of cell fate decisions underlies health and disease. To identify molecules that are active or regulated during a decision, and not before or after, the decision time point is crucial. However, cell fate markers are usually delayed and the time of decision therefore unknown. Fortunately, dividing cells induce temporal correlations in their progeny, which allow for retrospective inference of the decision time point. We present a computational method to infer decision time points from correlated marker signals in genealogies and apply it to differentiating hematopoietic stem cells. We find that myeloid lineage decisions happen generations before lineage marker onsets. Inferred decision time points are in agreement with data from colony assay experiments. The levels of the myeloid transcription factor PU.1 do not change during, but long after the predicted lineage decision event, indicating that the PU.1/GATA1 toggle switch paradigm cannot explain the initiation of early myeloid lineage choice.

[1] Institute of Computational Biology, Helmholtz Zentrum München, 85764 Neuherberg, Germany. [2] Department of Biosystems Science and Engineering (D-BSSE), ETH Zurich, 4058 Basel, Switzerland. [3] Department of Mathematics, Technische Universität München, 85748 Garching, Germany. [4] Present address: Institute for Systems Biology, 401 Terry Ave N, Seattle, WA 98109, USA. Correspondence and requests for materials should be addressed to T.S. (email: timm.schroeder@bsse.ethz.ch) or to C.M. (email: carsten.marr@helmholtz-muenchen.de)

Tightly controlled and correctly timed cell fate decisions are crucial for the development and maintenance of any healthy organism. Understanding their molecular control is therefore essential for basic biological research and the development of future therapies. However, the identification of the exact time point when a cell fate decision happens is often impossible, since the emergence of an observable signal is usually delayed from the fate decision itself. If we can only observe the delayed signal, but not the actual decision-making process, factors that influence the decision remain unidentified.

Consider a cellular process where an unobservable event (e.g., a cell fate decision) leads to an observable phenotypic signal (e.g., a morphological change or the onset of a lineage marker) with a delay in time. From just observing the signal in non-dividing cells, one cannot infer the true time point of the unobserved event as the delay is typically unknown (Fig. 1a). However, if cells divide during the delay, this induces correlated signals in related cells (e.g., two sisters or four cousins, etc.). These correlated signals carry information about the length of the delay and hence about the timing of the unobserved event: for example, a delay of 1–2 generations causes correlated readouts in sister and cousin cells and suggests a decision in the mother or grandmother generation (see Fig. 1b).

Due to recent advances in time-lapse imaging and single-cell tracking[1–4], it is now possible to obtain large genealogies of single cells and observe correlated signals. For example in yeast, sister cells switch gene expression of a simple regulatory circuit in a correlated fashion[5]. In mammalian hematopoiesis, differentiation is typically read out via the expression of a lineage specific differentiation marker[6,7], e.g., the CD16/32 membrane receptor in the myeloid branch of hematopoiesis (Fig. 2a). However, these markers report the lineage decision only indirectly, because their expression is a delayed downstream consequence of a former unobserved event (Fig. 2b). Here, we parametrize the decision process and the marker delay in a computational model that combines a memory-less decision process and stochastic gene expression mimicking marker delay with graphical models and dynamic programming to cope with the computational complexity of genealogies. This allows us to calculate the probabilities of different decision scenarios (called hidden trees, Fig. 2c) and determine the most likely time point of the unobserved lineage decision.

We apply this method to a dataset of differentiating hematopoietic stem cell genealogies with annotated lineage marker onsets and find that myeloid/megakaryocytic-erythroid lineage decision happens several generations earlier than reported by lineage markers and that the dynamics of PU.1 during the lineage decision is inconsistent with a PU.1-based toggle switch driving the lineage decision.

## Results

### Prediction of an early cell fate decision in hematopoiesis.
One hematopoietic lineage decision is the choice of hematopoietic stem and progenitor cells (HSPCs) between the megakaryocytic-erythroid (MegE) and the granulocyte-macrophage (GM) lineage[7]. The mutually exclusive expression of the transcription factors PU.1 and GATA1 in mature GM and MegE cells, respectively (see e.g.,[8] for an overview), and their mutual binding and cross-antagonism inspired toggle switch models that predict transcription factor dynamics before and during this decision[9–14]. These models assume the switch to one of the cross-antagonistic transcription factors to precede and induce GM vs. MegE lineage choice, and serve as the de facto paradigm of binary cell fate choice on a molecular level[15]. However, since the exact timing of GM vs. MegE lineage choice remains unknown it is impossible to

quantify the dynamics of PU.1 and GATA1 immediately before and during the actual lineage decision making.

To identify the time windows of this HSPC lineage decision making, and to compare it to the dynamics of PU.1 and GATA1 regulation, we used a dataset of sorted murine HSPCs, where endogenous PU.1 and GATA1 proteins are tagged with yellow and red fluorescent proteins, respectively (see Hoppe et al.[7] for experimental details). Over 10,000 single cells have been tracked and quantified, generating cellular genealogies up to 12 generations deep. Definite GM lineage commitment is detected via CD16/32 onset using in-culture antibody staining (i.e., a fluorescent CD16/32 antibody is present in the medium and accumulates on cells that express CD16/32 on the membrane[16,17], Fig. 2a). MegE lineage commitment is read out via GATA1–mCherry upregulation. We analyze 54 GM-fated and 20 MegE-fated genealogies from three independent experiments (see Fig. 3a, b).

To infer the time point of lineage choice, i.e., the time when a HSPC loses multipotency and commits towards the GM- or MegE-lineage, we fit our computational model to the observed genealogies by maximizing the likelihood of the data with respect to the model parameters (see Methods). For each individual genealogy (from now on also called "tree"), we calculate its likelihood given model parameters by summing up overall possible scenarios of differentiation, decomposing each tree into subtrees, and calculating the probability for each subtree using a graphical model (see Methods and Fig. 2d). After extensive testing on synthetic data (Supplementary Notes 1-2), we use our computational method to predict the most likely time point of lineage choice based only on the temporal correlations in related cells of CD16/32 or GATA1 onsets, respectively. The model decomposes the onset distribution into a differentiation probability (Fig. 3c) and a lineage decision marker delay distribution (Fig. 3d) to fit the observed marker onset distributions (Fig. 3e) and the observed correlation patterns. As shown in Fig. 3c, the

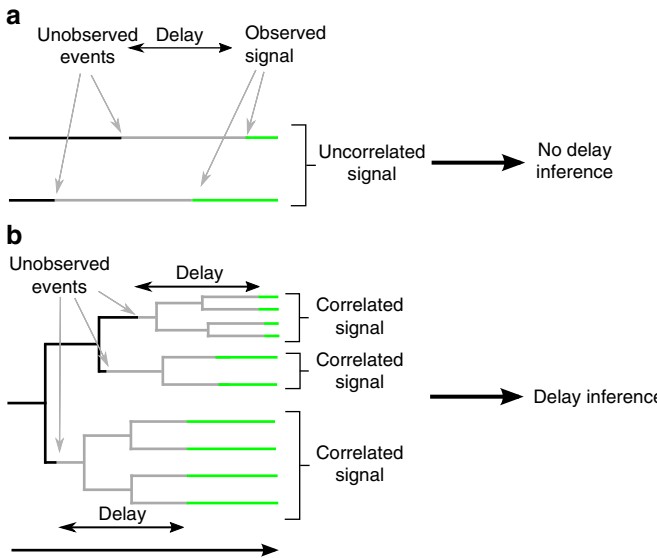

**Fig. 1** Correlated signals in genealogies allow to reconstruct the preceding unobserved events and infer the associated delay. **a** From uncorrelated observed signals (green), the time point of the preceding unobserved event (or likewise, the length of the delay) cannot be inferred (two samples shown). **b** In contrast, if cells divide during the delay process, signals become correlated and allow inferring the time points of the unobserved events

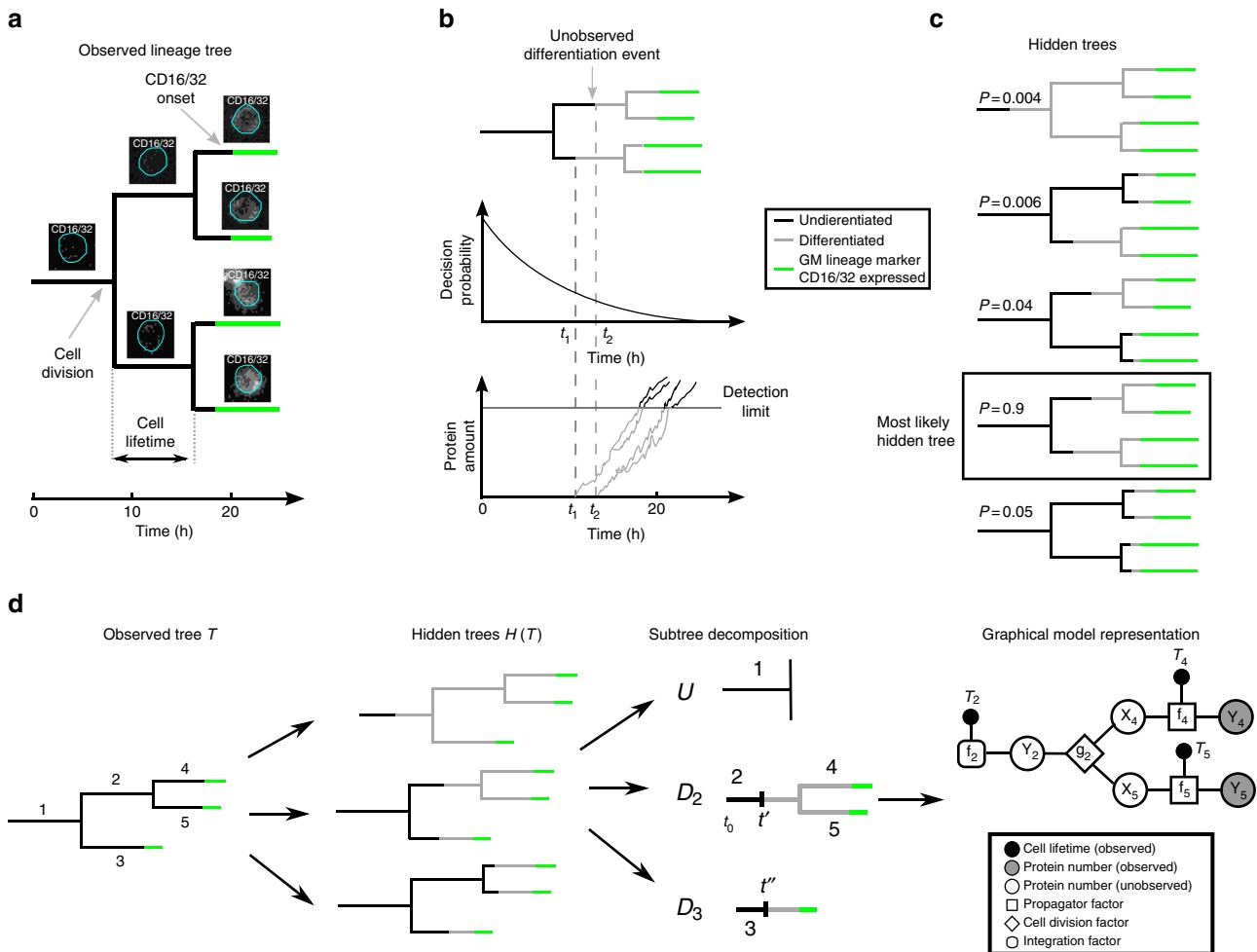

**Fig. 2** Correlated lineage marker onsets of differentiating HSPCs in time-lapse microscopy allow to infer the time point of lineage choice. **a** The tracking of dividing single cells in a time-lapse experiment gives rise to a genealogy. Expression of the lineage marker CD16/32 (indicated in green) is detected along the genealogy. **b** A simple differentiation model with an exponentially decreasing differentiation probability and a delay due to stochastic gene expression up to a detection limit induces marker correlations. After the first division, both cells independently differentiate at time $t_1$ and $t_2$ and start expressing the marker, but divide before reaching the detection threshold (gray line). Inheriting the state of the mother, the daughter cells will reach the threshold at similar, but due to stochasticity in gene expression, not identical times. **c** As the underlying dynamics are unknown, the observed marker onsets in a genealogy can originate from different possible differentiation scenarios termed hidden trees. The hidden tree with the highest probability is the most likely scenario of differentiation. **d** For one observed genealogy $T$, all hidden trees $H \in H(T)$ are constructed. A particular hidden tree can be decomposed into a single tree $U$ that contains only undifferentiated cells, and the set of subtrees $D_i$ whose roots are differentiating at unknown time points ($t'$, $t''$). To obtain its likelihood, each subtree $D_i$ is represented as a graphical model and message passing is performed (see Methods)

estimated differentiation probability is exponentially decreasing with time. The majority (74%) of predicted lineage decisions happen already in the first or second generation of the genealogies (Supplementary Figure 1). While the tracked generations are only relative to the start of the experiment, HSPCs had just been freshly sorted and had been kept at 4 °C from harvesting of bone marrow until shortly before the start of the imaging experiment, thus most likely preventing cellular decision making during HSPC preparation. Such early differentiation is surprising as the established lineage markers CD16/32 and GATA1 can only be detected after many days in culture[7]. Notably, the predictions of lineage decision time points are unchanged for moderate measurement noise (up to one cell cycle length) in the annotated onsets (see Supplementary Note 5). The delay between the unexpected early differentiation and the onsets of the lineage marker was on average 78 h for GM and 54 h for MegE (Fig. 3d), with cell cycle lengths of 12 ± 5 h (for further characterization of the delay process and its induced correlations, see Supplementary Note 6).

Interestingly, the differentiation probability distributions are almost identical for GM and MegE genealogies (Fig. 3c) even though they were estimated independently from different trees. This suggests a mechanism where a process common to both lineages determines the timing of differentiation, while the dynamics of lineage marker expression as a consequence of differentiation are distinct for both lineages.

Next, we validate our finding of early differentiation events using data from independent colony assay experiments of sorted HSPCs, performed in the same experimental conditions (Hoppe et al.[7]). These colony assays allow to read out the amount of pure GM-, pure MegE-, and mixed GMMegE (containing all lineages) colonies formed from single HSPCs after 10 days of culture. While the differentiation distribution $\Phi(t)$ cannot be measured directly, it leaves a distinct fingerprint in these frequencies: if lineage decisions happen early, and thus in few cells within the colony, mostly pure GM or pure MegE colonies will emerge, and GMMegE colonies will be rare. In contrast, if decisions happen late and thus independently in many cells within the colony, mostly GMMegE colonies will

emerge and pure GM or pure MegE colonies will be rare. This intuition can be formalized in a mathematical branching process model (see Methods and Marr et al.[18]), which predicts the proportions of GM, MegE, and GMMegE colonies for a given differentiation probability.

When supplied with the differentiation probability $\Phi(t)$ in Fig. 3c (estimated with our tree inference algorithm from time lapse data), the branching process model faithfully predicts the experimentally observed colony assay frequencies (see Fig. 3f, g). In particular, we are able to correctly predict the large frequency

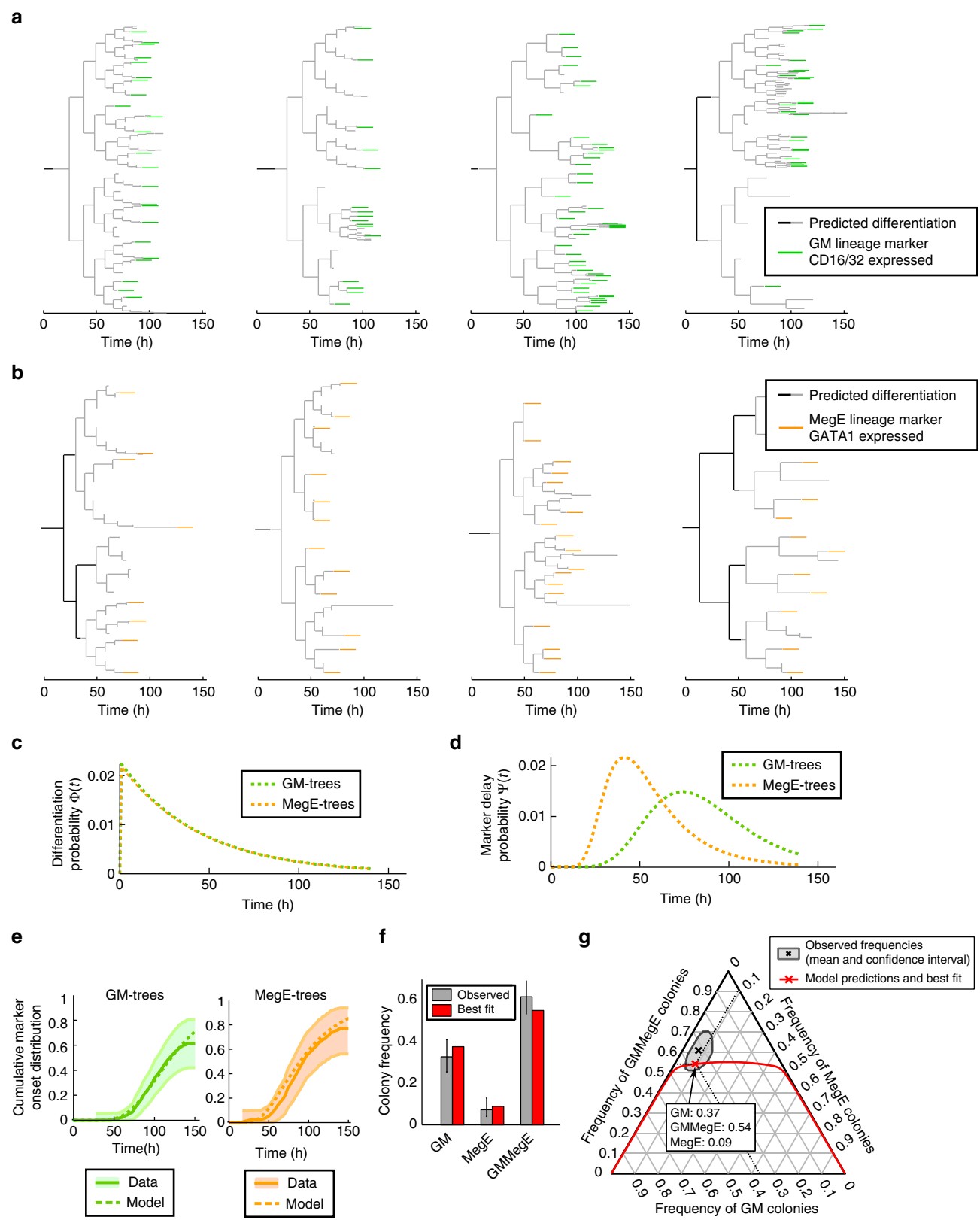

of observed GMMegE colonies (60 ± 7%), even though GMMegE genealogies were not used to estimate the differentiation probability with our tree inference algorithm (Fig. 3f, g). Note that GMMegE genealogies are rare in the time lapse dataset of Hoppe et al.[7], due to the tracking strategy applied, where trees are often only partially tracked. The few observed ones (see Supplementary Figure 18) are indeed consistent with early differentiation events.

**PU.1 dynamics at the predicted time point of lineage choice**. To investigate if PU.1 and GATA1 are the determinants or only a consequence of these HSPC lineage decisions, we analyzed the dynamics of endogenous PU.1 levels quantified from PU.1-eYFP fluorescence for each tracked cell and time point in the genealogies (Fig. 4a; for details, see Hoppe et al.[7]). In a typical branch of an HSPC genealogy, both the number of PU.1-eYFP proteins and the PU.1-eYFP concentration (intensity/cell area) rise before CD16/32 onset (Fig. 4a). This indeed matches the expectations from a toggle switch model including PU.1[9–11]: initially balanced, the switch tilts in favor of PU.1, which gets upregulated and leads to commitment towards the GM lineage, signified by delayed CD16/32 expression. However, it is impossible to tell a priori if PU.1 upregulation induces the lineage decision or if it is a downstream consequence of an earlier lineage choice.

To that end, we use the predicted lineage choice time point from our model and compare it to the time point of PU.1 upregulation. We find no significant difference in PU.1-eYFP production, quantified by estimating the slope of the PU.1-eYFP concentration (red lines in Fig. 4a) in cell generations before, at, or directly after the identified lineage choice time point, ($p = 0.25$ and $p = 0.15$, rank-sum test, see Fig. 4b). In contrast, PU.1-eYFP production is significantly higher in later cells with CD16/32 onset (Fig. 4b). Similarly, when inferring lineage choice time points in 20 MegE-fated genealogies based on correlated onsets of GATA1 expression, we find that PU.1-eYFP production does not change in cells before, at and directly after the predicted time point (Fig. 4c). These results are robust across three independent experiments (see Supplementary Note 4).

Now we compare these findings to a model where a toggle switch involving PU.1 drives cell differentiation. We implemented a popular toggle switch model that is thought to drive binary lineage decision composed of two mutually repressing transcription factors (Fig. 4d inset; see Supplementary Note 3 for model details)[9–14,19]. This model exhibits three stable states (Fig. 4d): The state where both proteins are expressed at similar levels is associated with a progenitor cell. In the two other states, one of the two proteins is strongly upregulated, thereby repressing the other, representing two mutually exclusive differentiated lineages. Differentiation initiation occurs via noise driven transitions from

the progenitor to one of the differentiated states. Using Gillespie's algorithm[20], we simulate genealogies from this toggle switch model starting from single cells in the undifferentiated state. Eventually this cell or its progeny will leave the progenitor state and proceed to one of the differentiated states, turning on marker expression. We now assume the underlying transcription factor dynamics to be unobserved and infer the putative differentiation time points from solely the correlated marker onsets in this synthetic dataset with our method. We find that in our synthetic dataset, the time point of predicted differentiation is identical to the time point where the toggle switch tilts (Fig. 4e). Here, the initial balance between the two factors is broken, one is upregulated while the other is downregulated in the predicted cells. Quantifying PU.1 production as in Fig. 4b, c, we find significant change between cell generations before and at the predicted differentiation time point, both for cells heading towards the GM-lineage (PU.1 upregulation, Fig. 4f) and towards the MegE lineage (PU.1 downregulation Fig. 4g). Similar results are found for different parameterizations and more complex models of the toggle switch (Supplementary Figures 12, 13). Our method is thus able to correctly predict differentiation events driven by a genetic toggle switch (Fig. 4e) and detect the cells where the involved transcription factors are differentially regulated (Fig. 4f, g) on synthetic data.

This approach shows that the experimentally observed marker onsets and PU.1 dynamics are inconsistent with a toggle switch involving PU.1, which initiates the lineage choice: If PU.1 was directly involved in the GM/MegE lineage decision, we would detect up- or downregulation in cells at the predicted time of lineage choice (compare Fig. 4b, f, as well as Fig. 4c, g). Thus, while PU.1's importance in the execution of GM/MegE programs is undoubted[21] (and demonstrated by knockout experiments[22,23]), it is not the initiator of lineage choice but rather an effector that locks down the chosen lineage.

## Discussion
The analysis of tree-structured data has a long history in the field of phylogenetics[24,25]. Here, the main challenge is to reconstruct a single unobserved sequence evolution tree using a stochastic model of nucleotide substitution[26] and observed sequences at the leaves of the tree. In contrast, we directly observe the stem cell genealogies, estimate model parameters from multiple trees, and use a complex stochastic model which makes parameter inference challenging. Due to the data structure and the lack of available tools, genealogies typically have been studied using summary statistics[18,27–30]. Modeling has only rarely been used to gain mechanistic understanding about the observations[31–33]. These approaches rely on a simple Markov model of state changes (akin to our differentiation process) and assume that this state change is readily observable. The key difference in our approach is that we allow for a delayed observation of the

**Fig. 3** Prediction of early decisions in GM and MegE lineages with long dissimilar delay. **a**, **b** Four exemplary GM-fated (**a**) and MegE-fated (**b**) genealogies from the dataset used to infer time points of lineage choice. **c** The estimated differentiation distributions $\Phi$ (and hence the differentiation rate $\lambda$) are almost identical between GM- and MegE-fated genealogies. **d** The estimated delay distribution $\psi$ show differences in the delay process between GM- and MegE-fated genealogies. **e** Combining the estimated $\Phi$ and $\psi$ into the estimated cumulative marker onset distribution (dashed lines) fits the observed cumulative marker onset distributions (solid lines: mean, shaded area: 95% confidence intervals) in GM (green) and MegE (orange) genealogies. **f** A branching process model supplied with the estimated differentiation distributions $\Phi$ correctly predicts observed colony assay frequencies ($n = 3$ experiments, errorbars indicate 95% confidence intervals). **g** In the space of all possible frequencies of GM, MegE, and GMMegE colonies (adding up to 1), the observed colony assay data occupies is a single point (black) and an associated confidence interval (gray). Varying its only free parameter ($p_{GM}$, see Methods), the branching process model traces a line (red) in frequency space and intersects the confidence region of the data. The frequencies of the best fit are marked with a red cross. While this model has one free parameter, its predictions with respect to GMMegE frequencies are almost independent of that parameter: the predicted frequency of GMMegE colonies is constrained to the [0.5, 0.55] interval unless either one of the non-GMMegE colony types is completely absent (left and right borders in Fig. 3G)

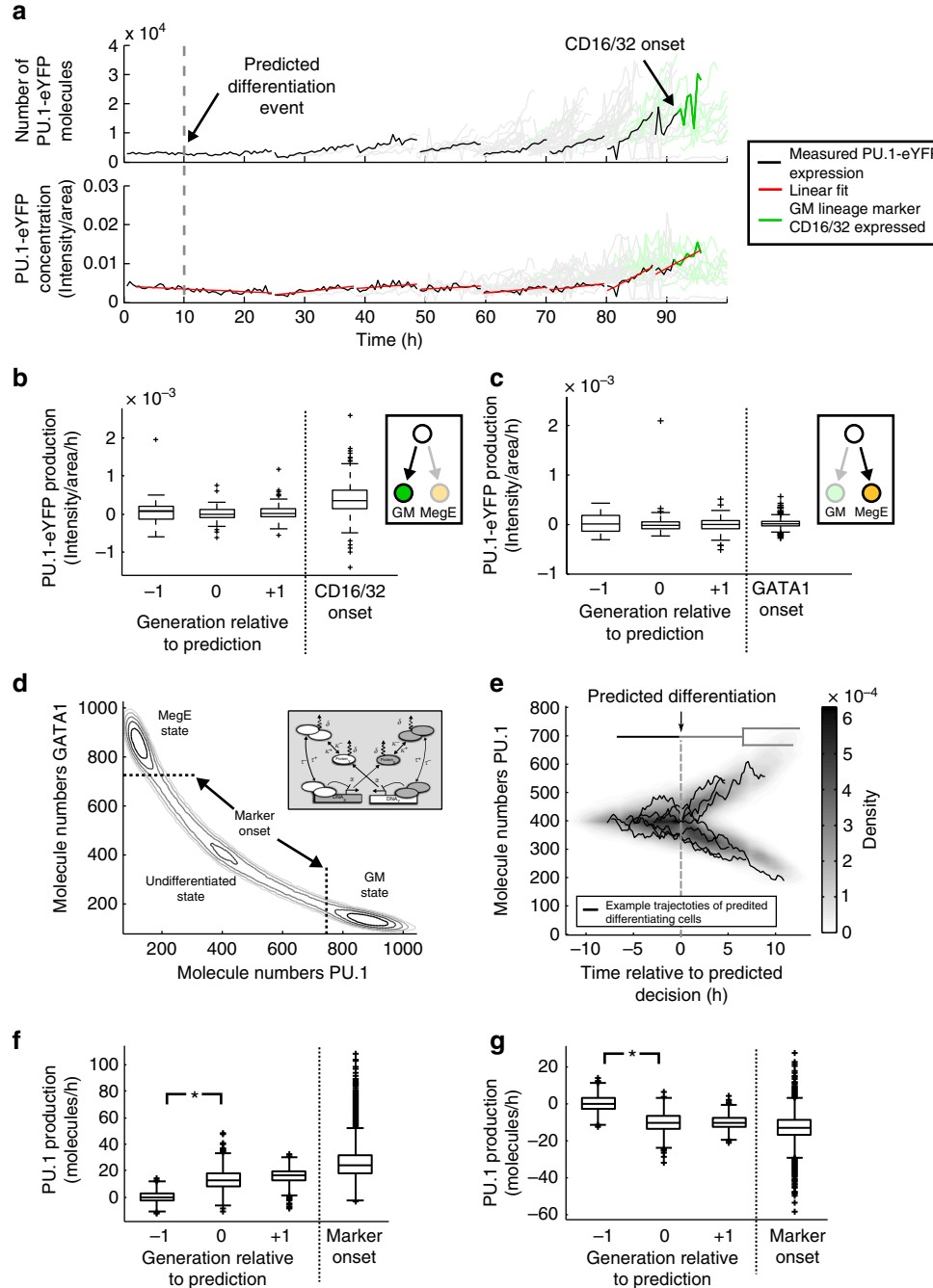

**Fig. 4** PU.1 expression in HSPCs is incompatible with simulated data from the toggle switch model. **a** PU.1-eYFP expression in terms of molecule numbers and concentrations. A single branch (black) of a full genealogy (gray) is highlighted that has a CD16/32 onset after seven rounds of division (green). PU.1-eYFP concentration is fitted with a linear model in each cell (red lines, only shown for the highlighted branch). **b** For GM-fated genealogies, PU.1-eYFP production (slope of the linear fit in b) is comparable in cells one generation before (−1), at (0), and one generation after (+1) the predicted lineage choice ($p = 0.25$ and $p = 0.15$, rank-sum test). In cells with CD16/32 onset, PU.1-eYFP production is increased. Boxplots indicate median, lower and upper quartile, whiskers display 1.5 × interquartile range. **c** For MegE-fated genealogies, PU.1-eYFP production is unchanged at the predicted lineage choice time point ($p = 0.54$ and $p = 0.79$, rank-sum test). Additionally, no downregulation of PU.1-eYFP is observed at the onset of the MegE lineage marker GATA1. **d** A toggle switch model of lineage choice (inset and Supplementary Note 3) gives rise to one undifferentiated (central) and two differentiated states (upper left, lower right) that can be identified as wells in the quasi-potential ($-\log(P)$) of the system. We define a cell to be lineage marker positive once it enters the basin of attraction of a differentiated state (dashed lines). **e–g** In genealogies simulated from a PU.1/GATA1 toggle switch model, a clear change in PU.1 production is observed at the predicted lineage choice time point. **e** The predicted lineage choice ($t = 0$, gray dashed line) coincides with the divergence of the toggle switch dynamics in single cells. Single trajectories (black lines), as well as the density across all predicted cells (color map) are shown. **f** In simulated GM-fated genealogies, the fitted slope of PU.1 abundance in cells predicted to differentiate (relative generation 0) is considerably higher as compared to undifferentiated cells one generation before ($p < 10^{-16}$, rank-sum test). The upregulation is directly linked to the in-silico lineage choice implemented by the toggle switch. **g** Similarly, in simulated MegE-fated genealogies, a downregulation of PU.1 (negative PU.1 production) starts in cells predicted to differentiate ($p < 10^{-16}$, rank-sum test)

underlying state change, detectable only several generations later. These long-range correlations are not accounted for by previous models.

Our model of differentiation and delay only approximates the underlying biological process. For example, we assumed that the differentiation rate is only time-dependent, whereas differentiation is likely to depend on other external factors, e.g., spatial interactions between cells and their microenvironment[34]. However, in our experiments, the high motility of blood progenitors results in fast mixing of cells and the impact of spatial interactions is presumably small (see Supplementary Figure 3). As time-lapse microscopy allows observing the spatial arrangement of cells, those effects can in principle be incorporated into the differentiation rate[35].

We modeled the marker delay as a simple stochastic gene expression due to a lack of knowledge about internal processes. Typical gene expression parameters[36] and reasonable detection limits would only allow for short delays in the range of hours. Correlations across multiple generations (as shown in Fig. 1a) however, cannot be explained by simple mechanisms, but are more likely caused by cascades in the underlying gene regulatory network that trigger differentiation. Our model can approximate such complex delay processes, e.g. via cascades of genes appropriately (see Supplementary Note 2-3).

A delay between the time point of lineage decision and the onset of lineage specific markers was expected, and some supporting data has been published. Paul et al.[37] recently found a population of cells with GM-like transcriptional profiles but without CD16/32 expression. In our setting, this corresponds to cells that are located downstream of a differentiation decision in a genealogy but are still negative for CD16/32 (gray cells in Fig. 3a). Our data suggests that the delay until marker onset is much longer than expected. Furthermore, we could show that the PU.1/GATA1 toggle switch model, a paradigm of lineage choice in hematopoiesis is inconsistent with the observed marker onsets and inferred delays; the data shows that PU.1 expression only changes significantly several generations after the inferred time of lineage choice. This is in line with the interpretation of Velten et al.[38] suggesting that the differentiation–tree model of hematopoiesis with binary lineage decisions at branching points should be revised. In addition, it fits to the finding that lineage choice can be predicted before marker onset based on cell morphology and movement[30]. In contrast to Kueh et al.[39], who report a cell cycle elongation upon PU.1 upregulation in an LMPP-like population, we see a decrease in cell cycle lengths from the first generation to the second, and a stabilization afterwards at around 12 h (see Supplementary Figure 2). Importantly the cell cycle distributions are similar for GM- and MegE-annotated genealogies (see Supplementary Figure 2). The prolonged cell cycle in the generations 0 and 1 is most likely a result of stem cells gradually getting activated and starting to cycle when exposed to the media conditions of the experiment.

Finally, it is highly interesting, and as yet without any explanation, how such a long delay between lineage choice and marker onset can be encoded in eukaryotic cells. In bacteria Levine et al.[40] demonstrated how a system of feedback loops could induce delayed cell fate decisions over several generations. However, it is unknown if similar mechanisms could account for the much longer delays on the order of several days, as estimated from our data.

Provided its extendibility and generality, we are confident that our method is applicable to a wide range of cellular decision problems. For example, it has been described that treatment of differentiating embryonic stem cells leads to a highly synchronized, delayed lineage choice days later[17,41]. Along the same line, reprogramming somatic cells into iPS cells is believed to be a stochastic process[42,43], and e.g. analyzing the timing of reprogramming[44] might give insight into this complex procedure. Similarly, it is thought that tumorigenesis is the result of stochastic state transitions between cancer stem cells and non-tumorigenic cells while metastases are generated when cells randomly undergo an epithelial–mesenchymal transition, detach from the tumor and spread the cancer into other body parts[45]. Here, our method could be used to trace back to the tumor- or metastasis-initiating cells in suitable time-lapse in vitro experiments in order to investigate what triggered these initial events.

## Methods

**Model assumptions**. We introduce a computational method that, based on observed correlations, estimates a delay to obtain the true time point of the unobserved decision. Although our method is generally applicable to any decision process and associated delay that leads to correlated outcomes in tree-structured data, we focused on cellular differentiation. Time lapse microscopy combined with cell tracking and fluorescence signal quantification delivers genealogies of single cells with fate annotation typically read out via surface markers or cell morphology[46–49] (Fig. 2a). Each genealogy starts with a single stem cell at $t = t_0$ (the start of time lapse microscopy). During the experiment, the cell divides and gives rise to two daughter cells. These cells will later also divide, giving rise to further progeny. At time points $t > t_0$ the onsets of lineage markers are observed (green in Fig. 2a).

We propose that observed correlations in marker expression emerge because of a delay between the unobserved differentiation time point and the observed marker onset. According to this generic model, an observed tree $T$ can be explained by several scenarios that we call "hidden trees" $\mathcal{H}(T)$ (Fig. 2c). In order to infer the true time point of the lineage choice, one has to assign probabilities to these alternatives and predict the mostly likely hidden tree given the observed data.

Therefore, we propose a simple model of lineage choice and delay based on two assumptions:

i. Lineage choice is independent between cells: No internal information is passed from mother to daughter cell that has influence on the timing. Thus, the probability to differentiate must only depend on factors that are not inherited during cell division. In the following, we will assume that the probability to differentiate is a function of time (see below).

ii. The delay between lineage choice and marker onsets originates from a gene expression process that starts after the differentiation decision. The marker onset is detected once the amount of marker proteins in the cell crosses a certain threshold $x^\star$ (Fig. 2c). If the cell divides before the protein amount exceeds the detection limit, its daughter cells inherit the marker expression from their mother. As daughter cells inherit the state of their mother, they become correlated with respect to marker onset; if one daughter reaches the detection limit, the other daughter will likely do the same. Because gene expression is intrinsically stochastic, the dynamics of both cells will not be exactly identical[50].

**Differentiation process**. We define a rate $\lambda(t)$ so that $\lambda(t)dt$ is the probability that the lineage decision occurs in the interval $[t, t + dt]$ in a single cell, given that it has not occurred yet in the interval $[0, t)$. Note that in survival analysis, $\lambda$ is called the hazard rate[51].

Next, we define the overall distribution of decision times $\phi(t)$, that is, the probability density to observe a decision at time $t$ (known as event density in survival analysis). Both quantities are related via (see Supplementary Methods):

$$\phi(t) = \lambda(t) \exp\left(-\int_0^t d\tau\, \lambda(\tau)\right).$$

For example, if $\lambda(t) = \lambda$ is constant, the above equation yields $\phi(t) = \lambda \exp(-\lambda t)$, which is the probability density of an exponential distribution. Without loss of generality, but motivated by experimental observation[18], we assume that the differentiation rate is a linear function of time such that

$$\lambda(t) = a_0 + a_1 t. \tag{1}$$

This represents a first order approximation to a potentially complex but unknown differentiation rate. It allows more flexibility than a zeroth-order approximation ($\lambda(t) = a_0$) and is sufficient to encompass mechanistic models of lineage choice (see Results). From now on, we denote the parameters of the differentiation process as $\theta = (a_0, a_1)$ and write $\phi(t|\theta)$ to make the dependence on the parameters explicit.

**Delay process.** We model the marker delay as a stochastic gene expression process. Combining transcription and translation for simplicity, we obtain a birth–death process with two reactions, one producing a protein with rate $\alpha$ and the other removing a protein with rate $\gamma$ (for details, see Supplementary Methods). We are only interested in the dynamics of the system until the protein numbers exceed the detection threshold $x^\star$, where we assume that the marker can be observed. The delay process is characterized by the first passage time distribution $\psi_{x_0}(t)$, that is, the probability that the protein number crosses the threshold $x^\star$ for the first time at time $t$ starting with $x_0$ proteins initially, and the propagat or $P_{x->x'}(t)$, the probability to start a state $x$ and after time $t$ arrive at state $x'$. Both $\psi_{x_0}(t)$ and $P_{x->x'}(t)$ depend on the parameters $\eta = (\alpha, \gamma, x^\star)$ of the underlying model, but we have dropped this dependence for readability. We obtain $\psi_{x_0}(t)$ and $P_{x->x'}(t)$ by numerically solving the Master Equation of the associated stochastic process (see Supplementary Methods).

**Statistical inference.** Our goal is to estimate the parameters $(\theta, \eta)$ of the model from observed genealogies in order to predict lineage choice in a given tree. To that end, we derive the likelihood $L(T|\theta, \eta)$ of an observed tree T given the parameters, which is then optimized to find the maximum likelihood estimates $\hat{\theta}, \hat{\eta}$.

The entire process of differentiation and marker delay on genealogies has the Markov property; given the internal state in terms of $\phi$ and $\psi$ of some cell $i$ at time $t$, the subtree induced by this cell is independent of all other cells in the entire tree. This allows us to divide the problem into smaller subproblems, where we enumerate on a per cell basis all possibilities of differentiation events in an observed tree, which we termed "hidden trees" (see Fig. 1c).

The likelihood of the observed tree T given parameters $\theta$ and $\eta$ is the sum of likelihoods of the hidden trees H, because these are competing alternatives (Fig. 2d):

$$\mathcal{L}(T|\theta, \eta) = \sum_{H \in \mathcal{H}(T)} \mathcal{L}(H|\theta, \eta) \qquad (2)$$

To derive the likelihood of a single hidden tree H, we partition the hidden tree into various subtrees $D_i$ induced by the differentiating cells and a single tree U that only contains undifferentiated cells (Fig. 2d). Due to the Markov property, the likelihood factorizes:

$$\mathcal{L}(H|\theta, \eta) = \mathcal{L}(U|\theta) \prod_i \mathcal{L}(D_i|\theta, \eta) \qquad (3)$$

Note that the parameters $\theta$ also appear in the likelihoods for $D_i$ as the root of these subtrees is still undifferentiated for some unknown time (Supplementary Figure 6). The first term is readily computed from the decision process (Eq. 1) as the process generating it has no memory and factorizes across cells in U. The terms $L(D_i|\theta, \eta)$ are more difficult to obtain, as the delay process has memory and hence the individual cells of the subtree cannot be treated independently. Also, one has to account for the unknown time interval where the root of the subtree is still undifferentiated (see Supplementary Figure 6). We represent each tree $D_i$ as a factor graph (Fig. 2d and Supplementary Methods). The factor graph models the dynamics of the delay process on the tree structure, whose state is only known at the leaves of the tree, where an onset is observed. We use message passing to integrate out all unobserved variables in the graph and thereby obtain the likelihood $L(D_i|\theta, \eta)$[52]. The sum over H in Eq. 2 consists of a large number of terms (it is double exponential in the number of cells[53]), hence an explicit summation is prohibitive for larger trees. However, the sum can efficiently be evaluated using dynamic programming (see Supplementary Methods).

Using Eq. 2 and 3, we can now perform maximum likelihood estimation of the underlying model parameters $\theta, \eta$ given a set of observed trees $T_1, \dots T_n$:

$$\left(\hat{\theta}, \hat{\eta}\right) = \underset{\theta, \eta}{\mathrm{argmax}} \sum_{i=1}^{n} \log[\mathcal{L}(T_i|\theta, \eta)]. \qquad (4)$$

To solve the above optimization problem, we apply a standard multiple-restart (Latin Hypercube[54]) optimization routine. Having learned the parameters $\hat{\theta}, \hat{\eta}$ via Eq. 4, we predict differentiation times and cells in the genealogies. For an observed tree T, we select the most likely hidden tree $\hat{H}$ from the set of all possible hidden trees according to

$$\hat{H} = \underset{H \in \mathcal{H}(T)}{\mathrm{argmax}} \quad \mathcal{L}\left(H|\hat{\theta}, \hat{\eta}\right) \qquad (5)$$

$\hat{H}$ is calculated recursively to avoid enumerating the entire set $\mathcal{H}(T)$ (Supplementary Methods) and is used to predict which cells most likely

differentiated. Note that one can additionally obtain the k-most likely hidden trees and their corresponding likelihoods (see Supplementary Figure 7).

**Branching process model for colony assays.** To validate the estimated differentiation rate $\lambda$ (Fig. 3c), we utilize colony assay data of single sorted HSPCs done in the same experimental conditions as the genealogies. Single HSPCs are sorted into separate microwells and form colonies over ten days. These colonies are classified into three categories: GM-colonies, which contain only granulocytes and monocytes; MegE-colonies, which contain only megakaryoctes and erythrocytes; and GMMegE colonies, which contain cell from both the GM and the MegE lineage. The relative frequencies and confidence intervals of these three colony types over the course of ten days are reported in Extended Data Fig. 6c of Hoppe et al.[7]. We consider only the data from day ten where the colony assay frequencies have stabilized.

Intuitively, the colony assay frequencies depend on the differentiation rate: early lineage choice will increase the fraction of homogeneous (GM or MegE) colonies whereas late lineage choice will give rise to mostly GMMegE colonies. This intuition can be formalized into a mathematical model (see Marr et al.[18] for details). The model has two parameters: the differentiation rate $\lambda(t)$ (related to the differentiation probability $\Phi(t)$, see Supplementary Note 1) and the lineage probabilities $P_{GM}$ and $P_{MegE}$ ($P_{GM} + P_{MegE} = 1$) for a single cell to pick either one or the other lineage upon differentiation. With these two parameters, one can derive recursive equations for the probability of observing a GM, MegE, and GMMegE colony as a function of cell generations. The frequency of a GM-colony after N generations is:

$$F_{GM}(N) = f_{GM}(N, 1)$$

where

$$f_{GM}(N, i) = \begin{cases} \lambda(i)p_{GM} + (1 - \lambda(i))\left(f_{GM}(N - 1, i + 1)^2\right) & N > 0 \\ 0 & N = 0 \end{cases}$$

The recursion is to be understood as follows: to yield a homogeneous GM colony, either the founding cell of the colony must differentiate and choose the GM lineage (first term of the sum), or the founding cell does not differentiate, but both its daughters in turn form homogeneous GM colonies (second term in the sum). The variable $i$ in $f_{GM}(N, i)$ is a mere bookkeeping device that keeps track of the current generation (due to a generation dependent $\lambda$). Note that in the case of constant $\lambda(i) = \lambda$, the result from Marr et al.[18] is obtained. An analogous formula applies for $F_{MegE}(N)$ and by definition, $F_{GemM}(N) = 1 - F_{GM}(N) - F_{MegE}(N)$.

As the colony assay model operates in discrete time (cell generations), we discretize the continuous differentiation rate $\lambda(t)$ obtained from the genealogies as follows:

$$\lambda(i) = \mathbb{E}_{s_i, c_i}\left[1 - e^{-\int_{s_i}^{s_i + c_i} \lambda(\tau) d\tau}\right]$$

where we take the expectation with respect to the birth time $s_i$ of a cell in generation $i$ and the cell cycle time $c_i$ of a cell in generation $i$. The expression inside the expectation is the probability to differentiate in the time interval $[s_i, s_i + c_i]$. Here, we account for the fact that cells in generation 1 and 2 tend to have longer cell cycles then cells in subsequent generations. Hence the hazard in generation 1 and 2 is increased due to prolonged cell cycle. The distribution of $s_i, c_i$ is readily estimated from the tracked genealogies (see Supplementary Figure 2).

Using the extracted generation-wise differentiation rate in the colony assay model, the only remaining free parameter is $p_{GM}$ (since $p_{MegE} = 1 - p_{GM}$). Note that we cannot use the fraction of GM and MegE genealogies in the time lapse dataset as a surrogate for $p_{GM}$, since the genealogies are not guaranteed to be tracked unbiased; their proportions do not reflect the true underlying lineage probabilities.

A parameter sweep of $p_{GM}$ (but fixed $\lambda(t)$) creates a curve in the ($F_{GM}$, $F_{MegE}$, $F_{GemM}$) space (see Fig. 3f), each point on the curve corresponding to a particular choice of $p_{GM}$. As the curve intersects with the confidence interval of the observed colony assay frequencies, the model is capable of explaining the observed colony assay frequencies. The predicted frequencies of the best fit match the observed frequencies (Fig. 3f, g).

**Data availability.** The datasets analysed during the current study are available from the corresponding author on request.

**Code availability.** An implementation of the computational method is available at https://github.com/QSCD/tree-inference.

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

## Acknowledgements

We thank Florian Büttner, Felix Buggenthin, and Jan Hasenauer for helpful discussion on the manuscript and Rene Schoeffel for computational support. This work was supported by the German Science Foundation DFG (project "Inference of Differentiation Decision Times from Blood Stem Cell Genealogies" to CM and SPP 1356 to FJT) and by the SNF to T.S.

## Author contributions

T.S. and F.J.T. conceived the project. M.K.S. implemented the models and algorithms and analyzed the data with C.M. P.S.H. conducted the experiments with D.L. and K.D.K.,

provided experimental data and analyzed the time-lapse data. T.S. planned and supervised generation of experimental data. F.J.T. contributed to model design. T.S. and F.J.T. contributed to discussion of the method and the results and contributed to the manuscript. M.K.S. and C.M. designed the study and wrote the manuscript with T.S.

## Additional information

**Competing interests:** The authors declare no competing interests.

