## [Peer Review File · Nature Communications]

Reviewers' comments:

Reviewer #1 (Remarks to the Author):

Several recent works have shown how cell genealogies can help infer properties of biological processes. In this paper, Strasser and co-workers employ this approach to estimate the time point at which a cell fate decision is made, namely the differentiation of hematopoietic stem cells into either the megakaryocytic-erythroid or the granulocyte-macrophage lineages. The results obtained seemed to disprove the established view that this cell fate choice is regulated by a mutual inhibition switch affecting the transcription factors PU.1 and GATA1, which has become common knowledge and a case example in the field of cellular decision making in recent years. The paper is nicely written and may represent a nice example of the use of cell genealogies in understanding cellular decision making. However, I am concerned about the lack of an independent validation of the results shown in Fig. 3. While Fig. 4 would have served as validation if an "accepted fact" would have been reproduced, reporting that PU.1 does not change at the decision point might not be a new discovery, but simply a mis-validation of the results. I consider this a sufficiently large potential problem that should be addressed before publication.

I also have the following minor comments:

1.- Figure 1B intends to be a scheme showing how correlations among genealogies allows inferring decision time points, but the two decisions shown are simply translated in time within the same division window. It might be more helpful to the reader if the two events occurred at different branch depths, and lead to branch correlations of different sizes.

2.- At the beginning of page 3 it says: "Thus, while expression of PU.1 and GATA1 had been analyzed in HSPCs before and after lineage decision making, it remained impossible to quantify their dynamics immediately before and during the actual lineage decision-making." The first reference to decision making should be changed to some other term, since the authors want to emphasize that it is not the actual decision making. Otherwise the sentence reads somewhat incoherent.

3.- Can the authors explain a bit better, for the sake of non-expert readers like me, what "in-culture antibody staining" is? Can this method be applied while tracking the cells with microscopy?

4.- When referring to Figure 3C, the authors say: "Surprisingly, the majority (74%) of predicted differentiation decisions happen already in the first or second generation of the genealogies." Why is this surprising? Is $t=0$ something else than the beginning of cell tracking?

5.- It is extremely surprising that the differentiation probability distributions overlap so well in Figure 3B. Can the authors explain intuitively why this happens?

6.- In the Discussion section, the authors refer to "the high motility of blood progenitors" in their experiments. I think this should be quantified better, since sister cells could be correlated just by being close to each other. How motile are your cells?

7.- Also in the Discussion section, the authors say "Finally, it is highly interesting, and as yet without any explanation, how such a long delay between cell fate choice and marker onset can be encoded as precisely timed as we observed. Current candidate mechanisms for deterministic programming of a delayed response like transcription factor motives, chromatin modifications or feedback signaling do not fit to a delay of up to 7 generations while enabling the synchronized marker onset in related sisters within only minutes to hours." First, the delay is not so precisely timed, since the delay distribution is quite wide (Fig. 3D). Second, I would like to point out that

mechanisms exist in bacteria by which such multigeneration delay can be generated (see for instance Levine et al, PLoS Biol. 2012, 10:e1001252).

Reviewer #2 (Remarks to the Author):

The model that the authors present here addresses a vital aspect of developmental decision making and its measurement, and the arguments motivating this model are excellent. This work has the potential to become a widely applied classic. However, in the very abbreviated form as submitted, this manuscript leaves out critical details of validation that would be needed for readers (and potential users) to evaluate how well this model actually performs to locate the time windows of developmental lineage decisions. While the careful mathematical treatment in the Supplement is fine, the main text offers very limited discussion of its relationship to the underlying biological data and treatment of different sources of biological variance. It also leaves out discussion of other factors that could influence the relationship of the measured indices to the underlying developmental process, and how these might be taken into account. Therefore, some revisions are needed, and the authors are strongly encouraged to make use of the longer format permitted by Nature Communications.

1. None of the figures in the present version seem to compare actual data points on the same graph with the model predictions, except Fig. S8D. Most readers would expect to see some comparison like this between prediction, range of likely values in prediction, and observation +/- measurement error, and this should be included in the main figures.

2. Fig. S4 is particularly useful to illustrate the way the model works. Something like this could also be very helpfully included in an extended version of the main figures.

3. In an expanded version, it would also be good to include more characterization of the Meg/E lineages and more of the Meg/E representative pedigrees in the main figures, not just the GM pedigrees.

4. The inference that two sibling pairs do or do not share a pre-committed ancestor, ultimately a yes/no call, depends on determining whether the measured times of differentiation are correlated between sibling pairs as well as between siblings (e.g., in Fig. 2B,C). Therefore, it is important to articulate clearly to a broad audience how this "Boolean" determination is related to measured time variances, and how much measurement noise is taken into account in this call. Please explicitly discuss how tightly correlated two differentiation marker onset times need to be in order to be considered correlated. This may very well be embedded in the statistical distributions generated by the model, but for a general audience the issues of measurement error and whether inter- vs. intra-clonal variances are used for validation need to be explained directly.

5. A number of biological features have been reported to correlate with the three properties measured here, GATA-1 upregulation, CD16/32 upregulation, and PU.1 upregulation. For example, in a fetal liver precursor system, PU.1 upregulation from an LMPP-like population appears to be a result of cell cycle elongation (H. Y Kueh et al., Science, 2013, not cited here). There is no explicit discussion of cell cycle times here or their relationship to different fates or phenotypic changes. Is this different in the bone marrow system used here? If indeed there is any trend toward cell cycle length change in either major lineage, then in what direction(s) and where in the process of developmental delay does it occur?

6. What is the likelihood that all the input cells in the study populations are already committed either to a Meg/E fate or to a GM fate? If that were the case, then indeed the model would be correct that the two lineages share an early process, but the result would be less informative about the timing between the divergence point and the obvious differentiation output. An important consideration is whether there any significant proportion of lineages that have both

Meg/E and GM fates among their progeny, and if so, how cells within those lineages behave. If not, then the early steps in the differentiation delay are not measured and can't be compared here.

7. It's not clear why the authors initially treat upregulation of GATA-1, a transcription factor important for many later gene expression changes associated with at least two developmental lineages, as a differentiation "marker" equivalent to upregulation of CD16/32, a terminal differentiation gene regulated by PU.1 and multiple other myeloid-specific transcription factors. In that sense, the result in Fig. S8D is just to be expected even if there should be no difference between the lineages. If one compares times of upregulation of GATA-1 and PU.1 in the two branches, as more equivalent changes within a pathway, then does the delay seem more similar between the two major branches?

8. Ultimately PU.1 levels are lower in the Meg/E lineages than in the GM lineages. Published work from the Akashi and Nutt labs suggests that the difference is already there before final commitment to particular terminal fates. However, Fig. 4 seems to show equivalent and approximately zero PU.1 expression in the progenitors of both lineages. Is downregulation of PU.1 in the Meg/E lineages detectable in this system, and is there any predictive heterogeneity of PU.1-eYFP expression in the starting population? Alternatively, is low PU.1-eYFP below the threshold of detection? This is not explicitly discussed in the main text, and the supplementary figure title in Fig. S1 does not accurately describe that the figure shows only GATA-1 expression. If this is a major point of Hoppe et al., then a citation is fine but a clear statement should be included here.

Reviewer #1 (Remarks to the Author):

Several recent works have shown how cell genealogies can help infer properties of biological processes. In this paper, Strasser and co-workers employ this approach to estimate the time point at which a cell fate decision is made, namely the differentiation of hematopoietic stem cells into either the megakaryocytic-erythroid or the granulocyte-macrophage lineages. The results obtained seemed to disprove the established view that this cell fate choice is regulated by a mutual inhibition switch affecting the transcription factors PU.1 and GATA1, which has become common knowledge and a case example in the field of cellular decision making in recent years. The paper is nicely written and may represent a nice example of the use of cell genealogies in understanding cellular decision making. However, I am concerned about the lack of an independent validation of the results shown in Fig. 3.

We thank the reviewer for assessing our manuscript. We agree that validating our results is desirable. In the revised manuscript, we thus use data from independent colony assay experiments (see Hoppe et al., Nature, 2016, Extended Data Fig. 6c), where cells have been seeded and grown in clonal density and the clonal composition has been analyzed after 10 days, to challenge our estimated differentiation distribution. We find that a branching process faithfully predicts the experimentally observed colony assay frequencies (see new Figure 3F,G) when supplied with the differentiation distribution in Figure 3C, which has been estimated with our tree inference algorithm from time lapse data. In the revised version of our manuscript, we describe our validation approach in the main text as follows:

“Next, we validate our finding of early differentiation events using data from independent colony assay experiments of sorted HSPCs, performed in the same experimental conditions (Hoppe et al. [6]). These colony assays allow to read out the amount of pure GM-, pure MegE- and mixed GMMegE (containing all lineages) colonies formed from single HSPCs after 10 days of culture. While the differentiation distribution $\Phi(t)$ cannot be measured directly, it leaves a distinct fingerprint in these frequencies: If differentiation happens early, and thus in few cells within the colony, mostly pure GM or pure MegE colonies will emerge, and GMMegE colonies will be rare. In contrast, if differentiation happens late and thus independently in many cells within the colony, mostly GMMegE colonies will emerge and pure GM or pure MegE colonies will be rare. This intuition can be formalized in a mathematical branching process model (see Methods and Marr et al. [17]), which predicts the proportions of GM, MegE and GMMegE colonies for a given differentiation probability.

When supplied with the differentiation probability $\Phi(t)$ in Fig. 3C (estimated with our tree inference algorithm from time lapse data), the branching process model faithfully predicts the experimentally observed colony assay frequencies (see Fig. 3F,G). In particular, we are able to correctly predict the large frequency of observed GMMegE colonies ($60 \pm 7\%$), even though GMMegE genealogies were not used to estimate the differentiation probability with our tree inference algorithm (Fig. 3F,G). Note that GMMegE genealogies are rare in the time lapse data set of Hoppe et al. [6], due to the tracking strategy applied, where trees are often only partially tracked.”

Furthermore we describe the colony assay model in the methods section of the manuscript. In conclusion, the colony assay data validates our estimated model parameters and confirms our prediction of early differentiation in hematopoietic genealogies.

While Fig. 4 would have served as validation if an "accepted fact" would have been reproduced, reporting that PU.1 does not change at the decision point might not be a new discovery, but simply a mis-validation of the results. I consider this a sufficiently large potential problem that should be addressed before publication.

We would like to emphasize that while we are not able to prove the correctness of our decision point directly, we can show that the 'accepted fact' (Pu1 Gata1 cross-inhibition drives the decision) is incompatible with the observed data in Fig. 4. To that end, we compared the changes in PU.1 production as predicted from the 'accepted fact' model in Fig. 4C with the observed changes in Fig. 4B at the inferred decision point. While in the simulated data, PU.1 production changes when the differentiation decision happens, this is not observed in the HSPC data. In the revised version of our manuscript, we tried to clarify this by rewriting the caption of Fig. 4 which now reads:

"Figure 4. PU.1 expression in HSPCs is incompatible with simulated data from the toggle switch model."

Moreover, we believe that the validation of our approach with independent data (see point above) strongly suggests that the inferred decision time point are correct.

I also have the following minor comments:

1.- Figure 1B intends to be a scheme showing how correlations among genealogies allows inferring decision time points, but the two decisions shown are simply translated in time within the same division window. It might be more helpful to the reader if the two events occurred at different branch depths, and lead to branch correlations of different sizes.

We thank the reviewer for this remark and changed Fig. 1B accordingly. Now, the decisions happen at different generations and illustrate the occurrence of different correlations more clearly.

2.- At the beginning of page 3 it says: "Thus, while expression of PU.1 and GATA1 had been analyzed in HSPCs before and after lineage decision making, it remained impossible to quantify their dynamics immediately before and during the actual lineage decision-making." The first reference to decision making should be changed to some other term, since the authors want to emphasize that it is not the actual decision making. Otherwise the sentence reads somewhat incoherent.

Thanks! Indeed, the usage of terms has been incoherent and misleading here. We changed the sentence in the revised version of our manuscript, where we now write:

"However, since the exact timing of GM versus MegE lineage choice remains unknown, it is impossible to quantify the dynamics of PU.1 and GATA1 immediately before and during the actual lineage decision-making."

3.- Can the authors explain a bit better, for the sake of non-expert readers like me, what "in-culture antibody staining" is? Can this method be applied while tracking the cells with microscopy?

In-culture antibody staining is a technique that allows to quantify expression of cell surface markers (e.g. CD16/32) without further genetic modification of the mouse line (where one would for example have to insert a CD16/32 fluorescence reporter construct into the genome). Instead, one adds a fluorescent antibody of CD16/32 into the medium. Due to its low concentration, the antibody's fluorescence is not detectable unless it accumulates on cells expressing the corresponding antigen: As soon as a cell expresses CD16/32 on its surface, it will appear fluorescent and be annotated as CD16/32+. We changed the manuscript accordingly:

"Definite GM lineage commitment is detected via CD16/32 onset using in-culture antibody staining (i.e. a fluorescent CD16/32 antibody is present in the medium and accumulates on cells that express CD16/32 on the membrane [15], [16], Fig. 2A)"

4.- When referring to Figure 3C, the authors say: "Surprisingly, the majority (74%) of predicted differentiation decisions happen already in the first or second generation of the genealogies." Why is this surprising? Is $t=0$ something else than the beginning of cell tracking?

Indeed, $t=0$ is the beginning of the movie and the cell tracking. Before that, cells have been extracted from mice, FACS sorted, and cultured for a maximum of a few hours. The early differentiation decision was not expected by our biological partners, mainly due to the expression of differentiation markers only after many days in culture.

5.- It is extremely surprising that the differentiation probability distributions overlap so well in Figure 3B. Can the authors explain intuitively why this happens?

We briefly mentioned an explanation in the manuscript already, i.e. first a shared "escape process" (independent of the GM vs MegE choice), and then a second process which determines the actual lineage. We extended this discussion in the revised version:

"This suggests a mechanism where a process common to both lineages determines the timing of differentiation, and separate decisions for differentiation

and for lineage choice. A cell initially resides in a progenitor state upstream of both lineages. When the cells leave that state, the future fate (GM or MegE) is not yet determined. Only after a cell has definitely left the progenitor state, it will decide between the GM and MegE fate and the respective markers CD16/32 and GATA1 are then expressed with different kinetics."

6.- In the Discussion section, the authors refer to "the high motility of blood progenitors" in their experiments. I think this should be quantified better, since sister cells could be correlated just by being close to each other. How motile are your cells?

We now quantify the motility of our cells in the new Supplementary Figure S3. On average, cells move 224 microns per hour. The average cell diameter is 14 microns, for comparison. Furthermore, we quantified the distance between sister cells over time: Two sister cells start in close proximity after their mother's division and remain close by (median distance 25 micron) for about 1h (see Supplementary Figure S3B). However, they rapidly move apart afterwards (median distance 4h after division: 190 microns). Due to this fast movement of cells, it is unlikely that spatial effects, like local microenvironments, influence the differentiation decisions.

7.- Also in the Discussion section, the authors say "Finally, it is highly interesting, and as yet without any explanation, how such a long delay between cell fate choice and marker onset can be encoded as precisely timed as we observed. Current candidate mechanisms for deterministic programming of a delayed response like transcription factor motives, chromatin modifications or feedback signaling do not fit to a delay of up to 7 generations while enabling the synchronized marker onset in related sisters within only minutes to hours." First, the delay is not so precisely timed, since the delay distribution is quite wide (Fig. 3D). Second, I would like to point out that mechanisms exist in bacteria by which such multigeneration delay can be generated (see for instance Levine et al, PLoS Biol. 2012, 10:e1001252).

We thank the reviewer for this helpful remark. We included the reference and adapted the discussion, where we now write:

"In bacteria Levine et al. [38] demonstrated how a system of feedback loops could induce delayed cell fate decisions over several generations. However, it is yet unknown if similar mechanisms could account for the much longer delays estimated from our data (on the order of several days)."

Reviewer #2 (Remarks to the Author):

The model that the authors present here addresses a vital aspect of developmental decision making and its measurement, and the arguments motivating this model are excellent. This work has the potential to become a widely applied classic. However, in the very abbreviated form as submitted, this manuscript leaves out critical details of validation that would be needed for readers (and potential users) to evaluate how well this model actually performs to locate the time windows of developmental lineage decisions. While the careful mathematical treatment in the Supplement is fine, the main text offers very limited discussion of its relationship to the underlying biological data and treatment of different sources of biological variance. It also leaves out discussion of other factors that could influence the relationship of the measured indices to the underlying developmental process, and how these might be taken into account. Therefore, some revisions are needed, and the authors are strongly encouraged to make use of the longer format permitted by Nature Communications.

We thank the reviewer for the positive overall assessment of our work. We gladly address the issues mentioned below and adapted and extended our manuscript accordingly.

Most importantly, we now use data from independent colony assay experiments (see Hoppe et al., Nature, 2016, Extended Data Fig. 6c), where cells have been seeded and grown in clonal density, and the clonal composition has been analyzed after 10 days, to challenge our estimated differentiation distribution. We find that a branching process faithfully predicts the experimentally observed colony assay frequencies (see new Figure 3F) when supplied with the differentiation distribution in Figure 3D, which has been estimated with our tree inference algorithm from time lapse data. This independently validates our model estimated from the genealogies. See also our response to point 6 below.

1. None of the figures in the present version seem to compare actual data points on the same graph with the model predictions, except Fig. S8D. Most readers would expect to see some comparison like this between prediction, range of likely values in prediction, and observation +/- measurement error, and this should be included in the main figures.

We would like to note that a direct visual comparison of model and data is challenging in our case, since one has to compare simulated trees with observed trees instead of simple "datapoints". Following the suggestion of the reviewer, we now compare the observed with the estimated cumulative marker onsets distribution in Fig 3E.

2. Fig. S4 is particularly useful to illustrate the way the model works. Something like this could also be very helpfully included in an extended version of the main figures.

We thank the reviewer for this suggestion and revised the main figures substantially. Details of the model construction are now shown in Fig. 2D.

3. In an expanded version, it would also be good to include more characterization of the Meg/E lineages and more of the MegE representative pedigrees in the main figures, not just the GM pedigrees.

Following the reviewer's suggestion, we now include MegE pedigrees in the expanded version of Fig 3.

4. The inference that two sibling pairs do or do not share a pre-committed ancestor, ultimately a yes/no call, depends on determining whether the measured times of differentiation are correlated between sibling pairs as well as between siblings (e.g., in Fig. 2B,C). Therefore, it is important to articulate clearly to a broad audience how this "Boolean" determination is related to measured time variances, and how much measurement noise is taken into account in this call.

We agree that the effect of measurement noise in the timing of the marker onsets should be investigated. While the marker onsets were manually annotated by experts during cell tracking and independently verified (we automatically quantified the fluorescence signal of both lineage markers and inspected/corrected each annotated onset to match marker onset dynamics), a certain variability might confound these annotations. We therefore studied how robust the predictions are when we artificially perturb the onsets in time and added this analysis into the Supplementary Material, Section 6. In summary, the predictions of cell differentiation events are unchanged for moderate measurement noise, that is, if onset perturbations are smaller than the duration of one cell cycle. We mention this analysis in the main text, where we now write in the Result section:

"Notably, the predictions of cell differentiation events are unchanged for moderate measurement noise (up to one cell cycle length) in the annotated onsets (see Supplementary Material, section 6)."

For large amounts of measurement noise (such that onsets are frequently shifted into more distant ancestor or descendant cells) the predicted differentiation events shift accordingly. While this shift is to be expected, we do not believe that such large measurement noise is realistic for our system.

Please explicitly discuss how tightly correlated two differentiation marker onset times need to be in order to be considered correlated. This may very well be embedded in the statistical distributions generated by the model, but for a general audience the issues of measurement error and whether inter- vs. intra-clonal variances are used for validation need to be explained directly.

Indeed, the determination of the decision time is embedded in the model, in particular in the stochastic process associated with the delay. In general, it is difficult to find clear cutoffs in the timing of marker onsets that distinguish between related vs. unrelated events, especially since the delay spans several generations. The stochastic marker delay process defines a multivariate probability distribution over all possible marker onsets and their correlations, not only pairs of onsets. Whatever falls inside this probability distribution would be

considered an onset pattern that is consistent with the stochastic process and hence can originate from a single differentiation event.

To illustrate the correlation structure induced by this stochastic process, we added a whole section in the Supplementary Material (Section 7: Analysis of the delay-induced correlations) and a Supplementary Figure S15, where we study the correlations in pairs of marker onsets. We refer to this section in the main text:

"For further characterization of the delay process and its induced correlations, see Supplementary material section 7"

5. A number of biological features have been reported to correlate with the three properties measured here, GATA-1 upregulation, CD16/32 upregulation, and PU.1 upregulation. For example, in a fetal liver precursor system, PU.1 upregulation from an LMPP-like population appears to be a result of cell cycle elongation (H. Y Kueh et al., Science, 2013, not cited here). There is no explicit discussion of cell cycle times here or their relationship to different fates or phenotypic changes. Is this different in the bone marrow system used here? If indeed there is any trend toward cell cycle length change in either major lineage, then in what direction(s) and where in the process of developmental delay does it occur?

Following the reviewer's suggestion, we now analyze cell cycle changes in more detail. In the revised discussion we write:

"In contrast to Kueh et al. [37], who report a cell cycle elongation upon PU.1 upregulation in an LMPP-like population, we see a decrease in cell cycle lengths from the first generation to the second, and a stabilization afterwards at around 12h (see Supplementary Figure S2). Importantly the cell cycle distributions are similar for GM- and MegE-annotated genealogies (see Supplementary Figure S2). The prolonged cell cycle in the generations 0 and 1 is most likely a result of stem cells gradually getting activated and starting to cycle when exposed to the media conditions of the experiment."

6. What is the likelihood that all the input cells in the study populations are already committed either to a Meg/E fate or to a GM fate? If that were the case, then indeed the model would be correct that the two lineages share an early process, but the result would be less informative about the timing between the divergence point and the obvious differentiation output.

We thank the reviewer for raising this important question. In the revised version of our manuscript, we now explicitly estimate the fraction of differentiated input cells in our model in the new Section 3.5 in the Supplementary Material. We find that this fraction is negligible (<0.00005). This also in line with the colony assay experiments from Hoppe et al.: If a substantial fraction of input cells would already be pre-committed, one would not observe the large amount of mixed GMMegE colonies (60%), but mainly GM- or MegE-colonies.

An important consideration is whether there any significant proportion of lineages that have both Meg/E and GM fates among their progeny, and if so, how cells within those lineages behave. If not, then the early steps in the differentiation delay are not measured and can't be compared here.

The colony assay experiments from Hoppe et al., 2016 show that there exists a large fraction of colonies with mixed lineages (GMMegE colonies). However, within the time-lapse microscopy dataset, trees with both lineages (GMMegE trees) are very rare since often only few cells in a tree have been tracked until marker onset. Hence, in the inference algorithm, we do not consider GMMegE trees (the size of the set of hidden trees would become very challenging).

In the revised manuscript, we now utilize the colony assay data from Hoppe et al. as an independent validation of our model and show that the estimated differentiation probabilities (Fig. 3C) correctly predict the existence of 60% GMMegE colonies (see Fig. 3F and also response to reviewer 1). In the revised version of our manuscript, we describe our validation approach in the main text as follows:

"Next, we validate our finding of early differentiation events using data from independent colony assay experiments of sorted HSPCs, performed in the same experimental conditions (Hoppe et al. [6]). These colony assays allow to read out the amount of pure GM-, pure MegE- and mixed GMMegE (containing all lineages) colonies formed from single HSPCs after 10 days of culture. While the differentiation distribution $\Phi(t)$ cannot be measured directly, it leaves a distinct fingerprint in these frequencies: If differentiation happens early, and thus in few cells within the colony, mostly pure GM or pure MegE colonies will emerge, and GMMegE colonies will be rare. In contrast, if differentiation happens late and thus independently in many cells within the colony, mostly GMMegE colonies will emerge and pure GM or pure MegE colonies will be rare. This intuition can be formalized in a mathematical branching process model (see Methods and Marr et al. [17]), which predicts the proportions of GM, MegE and GMMegE colonies for a given differentiation probability.

When supplied with the differentiation probability $\Phi(t)$ in Fig. 3C (estimated with our tree inference algorithm from time lapse data), the branching process model faithfully predicts the experimentally observed colony assay frequencies (see Fig. 3F,G). In particular, we are able to correctly predict the large frequency of observed GMMegE colonies ($60 \pm 7\%$), even though GMMegE genealogies were not used to estimate the differentiation probability with our tree inference algorithm (Fig. 3F,G). Note that GMMegE genealogies are rare in the time lapse data set of Hoppe et al. [6], due to the tracking strategy applied, where trees are often only partially tracked."

7. It's not clear why the authors initially treat upregulation of GATA-1, a transcription factor important for many later gene expression changes associated with at least two developmental lineages, as a differentiation "marker" equivalent to upregulation of CD16/32, a terminal differentiation gene regulated by PU.1 and multiple other myeloid-specific transcription factors.

Regarding the lineage markers we follow Hoppe et al., where we showed that GATA1 is indeed a faithful marker of MegE lineage commitment, i.e. any cell

expressing GATA.1 in the experiment will either become a Megakaryocyte or an Erythrocyte. No other markers of the MegE lineage are available in the dataset.

In that sense, the result in Fig. S8D is just to be expected even if there should be no difference between the lineages.

We think that the reviewer refers to Fig. 3D (S8D is model simulation) and the fact that the two delay distributions are different for GM and MegE. We agree that it is not surprising to have different delay distributions for both lineages as they just depend on how "immediate" the respective marker is expressed upon the differentiation. GATA.1 being linked to the cell fate decision with less delay than the cell surface marker CD16/32 might be the reason for the longer delay in GM-trees (Fig. 3D). Notably, if GATA.1 would be a deciding factor (as proposed by the standard toggle switch model), the delay should be 0.

If one compares times of upregulation of GATA-1 and PU.1 in the two branches, as more equivalent changes within a pathway, then does the delay seem more similar between the two major branches?

Using PU.1 upregulation as a marker of GM-fate is challenging: (i) One would have to establish experimentally that PU.1 expression above a certain level guarantees cells to be committed to the GM lineage. For GATA.1, conveniently, this threshold is the detection limit (see Hoppe et al.), i.e. any cell expressing GATA1 at detectable levels will become a MegE cell. For PU.1 however there is no clear cutoff. While cells expressing high amounts of PU.1 are indeed committed to the GM fate (Hoppe et al., Fig. 2B), for intermediate levels of PU.1 there is no distinct lineage readout. In other words, PU.1 expression is predictive of future cell fate to a certain extent but far from perfect, while GATA1 expression is a perfect predictor of cell fate. (ii) While GATA1 expression is 'irreversible' (once expressed, GATA.1 is going up), PU.1 expression is fluctuating and a clear onset is not annotated (the examples in our Fig. 4a might be a bit misleading here, the non-stereotypic expression of PU.1 is more evident in Fig. 4b of Hoppe et al., (see below) making the suggested comparison practically impossible.

8. Ultimately PU.1 levels are lower in the Meg/E lineages than in the GM lineages. Published work from the Akashi and Nutt labs suggests that the difference is already there before final commitment to particular terminal fates. However, Fig. 4 seems to show equivalent and approximately zero PU.1 expression in the progenitors of both lineages.

We would like to note that Fig. 4B and 4D show PU.1 production, not PU.1 expression. PU.1 production is 0 in both lineages, meaning that PU.1 protein levels are on average constant in the generations around the predicted onset. The expression of PU.1 as measured by absolute numbers in Fig. 4A and by concentration in Fig. 4B is however non-zero.

Is downregulation of PU.1 in the Meg/E lineages detectable in this system, and is there any predictive heterogeneity of PU.1-eYFP expression in the starting population?

Yes. As shown in Hoppe et al., PU.1 is downregulated during MegE maturation, and often before onset of detectable GATA1 expression (but, as shown here, after MegE lineage choice). Interestingly, we find no significant difference in PU.1 expression in early generations, using the deep learning based cell fate predictor from Buggenthin et al. (Nature Methods, 2017). For the reviewer's convenience, we show Fig. 2d and 2e of this paper here:

Caption: (d) Concentration of PU.1-eYFP for unknown and latent cells in generations after the experiment start; they are subdivided into predicted GM (blue) and MegE (red) on the basis of their CNN-RNN lineage score for round 1 (see supplementary fig. 4 for rounds 2 and 3). The PU.1-eYFP concentration is significantly different in generations 2 ($P < 0.01$) and 3-8 ($P < 0.001$, unpaired Wilcoxon rank-sum test) after experiment start between the two predicted groups. Error bars extend to 1.5 times the interquartile range. (e) Significantly different PU.1-eYFP expression in generation 2 after experiment start is detected in 55 MegE (red) vs. 34 GM (blue) predicted cells.

Alternatively, is low PU.1-eYFP below the threshold of detection? This is not explicitly discussed in the main text, [...]

PU.1-YFP is detectable in all HSCs at the start of the experiment, and 'low' PU.1 levels are still above the detection threshold. We would again like to note that Figs. 4B, 4D and S1B show production, not expression.

[...] and the supplementary figure title in Fig. S1 does not accurately describe that the figure shows only GATA-1 expression. If this is a major point of Hoppe et al., then a citation is fine but a clear statement should be included here.

The title of the former Figure S1 was indeed misleading. Fig. S1A is now contained in main Fig. 3B. In the revised version of the manuscript, we removed Fig. S1B as it is not essential for PU.1 dynamics.

Reviewers' comments:

Reviewer #1 (Remarks to the Author):

The authors have addressed in my opinion all the issues raised. I would only suggest them to include in the text of the paper their response to my point #4, regarding the definition of the origin of time in their experiments. Even though the discussions with the reviewers might be made public upon publication of the paper, I don't know if the authors will decide to do so, and in any case it would be helpful for the readers to see this clarification in the text of the paper, in my opinion.

Reviewer #2 (Remarks to the Author):

This exciting paper is enhanced by the authors' responses to the previous review, and the points it makes come through much more clearly. The authors' strong data showing that the PU.1 increase only occurs in cells that have already chosen not to make MegE descendants is very important. Aside from a few minor typos, only a couple of issues remain problems for this reviewer.

1. The addition of the GMMegE colony data is extremely welcome. However, it is still not clear how the populations giving 60-70% GMMegE colonies relate to the actual populations tracked in these studies. The genealogies followed in Fig. 3A,B are only GM-fated or MegE-fated. If this is really what those lineages looked like, then it is possible that those lineage progenitors had already made their decisions before even entering the analysis. Where are the genealogy data from lineages that give both GM and MegE progeny?

It is not really appropriate to say that cells make the decision between GM and MegE fates two cell divisions into the culture if one cannot follow any examples of cells that had both GM and MegE progeny. This point is worth emphasizing because the properties of any lineages in which a single progenitor did give rise to both types of progeny would provide an extremely valuable reality check for the modeling of the differentiation choice time.

2. Throughout the text and the Supplement there is some ambiguity in the way the authors are using the term "differentiate". In some places this appears to refer to the choice of what kinds of progeny an HSPC will make. This is the sense in which I believe the authors usually use the term, and that is what I refer to in my previous comment. In other places, though, it appears to mean the process of maturation along a pathway that has already been selected (e.g. reaching the point where GM cells express CD16/32). It would be helpful to distinguish these first two meanings by calling them different things, e.g. "fate selection" and "maturation". However, possibly the authors even have a third meaning in mind, namely the time when a pre-chosen pathway is triggered to begin its execution. This would be relevant to the inferred trees in Figs. 3A and 3B despite their lack of any obvious lineage choice point. However, I don't think this triggering event can be interpreted as the same thing as lineage choice. It would be very helpful if the authors could clarify and standardize their usage.

If these points could be cleared up, this paper should have a large impact on thinking about hematopoiesis and the modeling of cell fate determination.

Reviewer #1 (Remarks to the Author):

The authors have addressed in my opinion all the issues raised. I would only suggest them to include in the text of the paper their response to my point #4, regarding the definition of the origin of time in their experiments. Even though the discussions with the reviewers might be made public upon publication of the paper, I don't know if the authors will decide to do so, and in any case it would be helpful for the readers to see this clarification in the text of the paper, in my opinion.

We thank the reviewer for raising this point. Earlier, our response to the point #4
When referring to Figure 3C, the authors say: "Surprisingly, the majority (74%) of predicted differentiation decisions happen already in the first or second generation of the genealogies." Why is this surprising? Is $t=0$ something else than the beginning of cell tracking?

was:

"Indeed, $t=0$ is the beginning of the movie and the cell tracking. Before that, cells have been extracted from mice, FACS sorted, and cultured for a maximum of a few hours. The early differentiation decision was not expected by our biological partners, mainly due to the expression of differentiation markers only after many days in culture."

We gladly add the following clarification to the results section of revised version of our manuscript:

"The majority (74%) of predicted lineage decisions happen already in the first or second generation of the genealogies (Supplementary Fig. 51). While the tracked generations are only relative to the start of the experiment, HSCs had just been freshly sorted and had been kept at 4°C from harvesting of bone marrow until shortly before the start of the imaging experiment, thus most likely preventing cellular decision making during HSC preparation. Such early differentiation is surprising as the established lineage markers CD16/32 and GATA1 can only be detected after many days in culture [7]."

Reviewer #2 (Remarks to the Author):

This exciting paper is enhanced by the authors' responses to the previous review, and the points it makes come through much more clearly. The authors' strong data showing that the P.U.I increase only occurs in cells that have already chosen not to make MegE descendants is very important. Aside from a few minor typos, only a couple of issues remain problems for this reviewer.

1. The addition of the GMMegE colony data is extremely welcome. However, it is still not clear how the populations giving 60-70% GMMegE colonies relate to the actual populations tracked in these studies. The genealogies followed in Fig. 3A,B are only GM-fated or MegE-fated. If this is really what those lineages looked like, then it is possible that those lineage progenitors had already made their decisions before even entering the analysis. Where are the genealogy data from lineages that give both GM and MegE progeny? It is not really appropriate to say that cells make the decision between GM and MegE fates two cell divisions into the culture if one cannot follow any examples of cells that had both GM and MegE progeny. This point is worth emphasizing because the properties of any lineages in which a single progenitor did give rise to both types of progeny would provide an extremely valuable reality check for the modeling of the differentiation choice time.

Indeed, the colony assays in Hoppe et al. (2016) have been performed in separate experiments from the time-lapse microscopy experiments, under identical culture conditions. The discrepancy between the observed fraction of these GMMegE colonies and the tracked GMMegE genealogies is surprising at first sight, but has the following two reasons: (i) The GMMegE fraction of the colony assay as reported in our revised manuscript was read out at a single late timepoint (after 10 days, see Supplementare Information in Hoppe et al. (2016)), in contrast to the time lapse experiments, where individual cells were followed by tracking. (ii) Due to the extreme effort required for single cell tracking (more than a full year of manual tracking) to generate the used data set, only few genealogies were generated where all cells are followed up to marker onset. In most genealogies, only few cells were followed up until the markers appears (see the newly incorporated supplementary Fig. 54, showing all trees of the data set), since this tracking strategy yielded sufficient information for the question at hand in Hoppe et al. (2016). Genealogies where e.g. even only a single cell, or two sister cells, or four niece cells were tracked and all turn on the GM marker would be classified as a pure GM genealogy.

Following the reviewer's comment, we confirm that the few tracked GMMegE genealogies can be used to challenge the estimated model. We find that these trees are consistent with the early differentiation events that the model learned from the pure GM- and MegE-genealogies. We added Supplementary Fig. 518 showing exemplary GMMegE trees and discuss them in the revised Supplementary Material, Section 8:

"Genealogies that give rise to both GM and MegE cells are rare in the dataset of [1] (17 across three experiments) due to the tracking strategy, and only scarcely tracked. Hence, they were not used in estimating the model parameters from the tracked genealogies. However, they provide a test case for our prediction of differentiating cells: The common ancestors

of a GM-cell and a MegE cell must have been bipotent and hence differentiation can only occur below these common ancestors. For our model, we estimated that differentiation should predominantly happen in the first five generations (Fig. S1). The observed GMMegE genealogies support this finding (see Fig. S18 for eight exemplary GMMegE genealogies): Most GMMegE genealogies are consistent with a scenario of differentiation in the generations 2-4, and only a single GMMegE genealogy (Fig. S18F) indicates differentiation as late as generation 6. If instead differentiation would predominantly occur late, one would not see the largely homogeneous (in terms of GM vs MegE) subtrees."

2. Throughout the text and the Supplement there is some ambiguity in the way the authors are using the term "differentiate". In some places this appears to refer to the choice of what kinds of progeny an HSPC will make. This is the sense in which I believe the authors usually use the term, and that is what I refer to in my previous comment. In other places, though, it appears to mean the process of maturation along a pathway that has already been selected (e.g. reaching the point where GM cells express CD16/32). It would be helpful to distinguish these first two meanings by calling them different things, e.g. "fate selection" and "maturation". However, possibly the authors even have a third meaning in mind, namely the time when a pre-chosen pathway is triggered to begin its execution. This would be relevant to the inferred trees in Figs. 3A and 3B despite their lack of any obvious lineage choice point. However, I don't think this triggering event can be interpreted as the same thing as lineage choice. It would be very helpful if the authors could clarify and standardize their usage.

We would like to thank the reviewer for raising this important point. We have thoroughly revised the nomenclature in our manuscript and the figures accordingly. We now use the general term 'differentiation' to describe the process of one cell type becoming another. 'Lineage choice' is used specifically for the time point when one lineage is chosen irreversibly (that is, in the experimental conditions used) over the other:

"To infer the time point of lineage choice, i.e. the time when a HSPC loses multipotency and commits towards the GM- or MegE-lineage, [...]"

Finally, the maturation process between lineage choice and the appearance of the lineage marker signal is termed delay throughout.

REVIEWERS' COMMENTS:

Reviewer #2 (Remarks to the Author):

I would like to thank the authors for their positive and informative responses to the previous reviews. I am happy to support publication of their exciting work at this point.